# Analysis of the Relationship between Personality Traits and Driving Stress Using a Non-Intrusive Wearable Device

**Wilhelm Daniel Scherz** [1,2,*,†] , **Victor Corcoba** [3,†] , **David Melendi** [3] , **Ralf Seepold** [1] ,
**Natividad Martínez Madrid** [4] and **Juan Antonio Ortega** [2]

1 Department of Computer Science, HTWG Konstanz—University of Applied Sciences, Alfred-Wachtel-Str. 8, 78462 Konstanz, Germany; ralf.seepold@htwg-konstanz.de
2 School of Computer Engineering, University of Seville, Avda. Reina Mercedes s/n, 41012 Seville, Spain; jortega@us.es
3 Department of Computer Science, University of Oviedo, 33003 Gijón, Spain; corcobavictor@uniovi.es (V.C.); melendi@uniovi.es (D.M.)
4 School of Informatics, Reutlingen University, Alteburgstr. 150, 72762 Reutlingen, Germany; natividad.martinez@reutlingen-university.de
* Correspondence: wscherz@htwg-konstanz.de; Tel.: +49-7531-206-298
† These authors contributed equally to this work.

**Abstract:** While driving, stress is caused by situations in which the driver estimates their ability to manage the driving demands as insufficient or loses the capability to handle the situation. This leads to increased numbers of driver mistakes and traffic violations. Additional stressing factors are time pressure, road conditions, or dislike for driving. Therefore, stress affects driver and road safety. Stress is classified into two categories depending on its duration and the effects on the body and psyche: short-term eustress and constantly present distress, which causes degenerative effects. In this work, we focus on distress. Wearable sensors are handy tools for collecting biosignals like heart rate, activity, etc. Easy installation and non-intrusive nature make them convenient for calculating stress. This study focuses on the investigation of stress and its implications. Specifically, the research conducts an analysis of stress within a select group of individuals from both Spain and Germany. The primary objective is to examine the influence of recognized psychological factors, including personality traits such as neuroticism, extroversion, psychoticism, stress and road safety. The estimation of stress levels was accomplished through the collection of physiological parameters (R-R intervals) using a Polar H10 chest strap. We observed that personality traits, such as extroversion, exhibited similar trends during relaxation, with an average heart rate 6% higher in Spain and 3% higher in Germany. However, while driving, introverts, on average, experienced more stress, with rates 4% and 1% lower than extroverts in Spain and Germany, respectively.

**Keywords:** revised eysenck personality questionnaire; EPQR-S; personality trait; stress; heart rate variability (HRV); driving simulator; neuroticism; extroversion; psychoticism

## 1. Introduction

According to the World Health Organization (WHO), an estimated 1.3 million individuals succumb annually as a consequence of road traffic accidents [1]. Furthermore, a substantial number, ranging from 20 to 50 million people, experience non-fatal accidents, often resulting in various forms of incapacitation. The consequences of these accidents extend beyond personal suffering, significantly impacting both individuals and the nation. This impact is evident through the substantial economic burdens arising from the expenses associated with medical treatment, the decreased productivity resulting from injuries, and the necessity for family members to take time off work to provide care to the injured parties. Notably, traffic accidents account for a considerable 3% of the nation's gross domestic product. It is worth highlighting that stress is a noteworthy factor contributing to

the occurrence of traffic accidents, given its adverse effects on the cognitive capabilities of drivers [2].

Stress, as defined by Selye [3], is a non-specific physiological response to a combination of external stressors and internal concerns. This complex phenomenon shows a robust correlation with cognitive abilities, including inattention, anxiety, memory and perceptual-motor performance. The field of road safety has been the subject of previous investigations, where researchers have explored the effects of driving-related stress [4]. Driving-related stress manifests itself when the driver perceives a mismatch between their driving abilities and the demands of the driving environment. This stress then manifests itself in the form of errors and an increased propensity to commit traffic offenses. Several contributing factors, such as time constraints, traffic congestion, road conditions and dislike of driving, combine to increase this specific strain [5].

Apart from the previously discussed variables, many researchers have focused on examining the impact of personality traits on driving behavior. According to a study [6], increased levels of workplace stress are associated with higher levels of anger-related traits and a greater likelihood of engaging in aggressive driving, which in turn increases the risk of being involved in traffic accidents. Furthermore, various personality traits, such as neuroticism, extroversion, and altruism, among others, have been recognized as significant factors in shaping an individual's driving behavior.

Individuals with high levels of neuroticism tend to experience greater levels of perceived stress compared to those who are extroverted and conscientious [7]. Such individuals may perceive events as threatening and have limited coping resources and self-regulation mechanisms [8,9]. Dahlen et al. [10] found that certain personality traits, including agreeableness, openness, and emotional stability, are relevant factors in predicting unsafe driving. Clarke et al. [11] conducted a meta-analytic review of the Big Five personality factors and their relationship to accident involvement. The study revealed that individuals with high levels of extroversion, low conscientiousness, and agreeableness are more likely to be involved in traffic accidents. The article by Riendeau et al. [12] evaluates the impact of personality traits on driving performance. The study reveals that neuroticism and extroversion are associated with hazardous driving behaviors, such as overspeeding, oncoming lane trespass, and collisions. On the other hand, conscientiousness has a positive effect on safe driving, but only in middle-aged drivers.

As seen above, there are many works in the literature where the effect of personality on driving behavior is analyzed. However, there is a lack of studies discussing how personality affects driving stress. On the other hand, these works do not consider cultural factors. These could influence the impact of personality on both driving behavior and stress. The objective of this paper is to see if there is a relationship between personality traits and the stress experienced while driving. We also want to test whether the degree to which personality traits influence driving stress depends on cultural differences between two countries, in this case, Germany and Spain, in line with the conclusions of Fishbein and Ayzen [13].

Our work postulates two hypotheses:

- suggests a significant relationship between personality traits and driving stress, aiming to investigate whether personality traits have a notable influence on the experienced stress while driving.
- states that the degree to which personality traits influence driving stress depends on cultural differences between Spain and Germany. This hypothesis aims to explore the potential influencing effects of cultural differences on the relationship between driving stress and personality traits, suggesting that the impact of each personality trait on driving stress may vary across different cultures.

We performed driving experiments with a set of 71 participants from Germany and Spain and the results have shown differences in conducting stress due to the different cultures. This study seeks to provide a deeper understanding of the complex relationships and influences between cultural factors, personality traits and stress, by validating the two hypotheses that have been postulated. The insights gained from this research can contribute

to the development and fine-tuning of semi-automated assistance systems, reducing errors that may occur due to stress. The study's practical applications may extend to improving driving safety and the overall effectiveness of assistance technologies.

The rest of the paper is organized as follows. The next section starts by introducing related work. Section 3 describes the methodology and statistical analysis. Section 4 shows the results of the analysis carried out, organized by country (Germany and Spain) and by personality dimension. Section 5 presents the discussion of the results and Section 6 exposes the conclusions.

## 2. Literature Review

Stress is a biological process mainly led by two chemicals: cortisol and noradrenaline [14]. The latter causes the body to increase heart rate and elevate blood pressure [14]. Thus, certain heart rate characteristics serve as indicators of stress levels. For example, the measurement of the fluctuations in the interval between consecutive heartbeats (known as heart rate variability—HRV [15]) is a reliable method to evaluate our autonomous nervous system in order to detect stress levels [16]. This relationship between stress and HRV is caused by the bidirectional communication between our nervous and immune systems [17–19].

There are different biomarkers and sensor technologies to assess HRV. Marques et al. [16] try to assess stress by collecting sweat and saliva and by measuring the HRV. The usage of these techniques allows the authors to assess stress by non-intrusive means. The authors emphasize the critical importance of employing non-stress-inducing methodologies (non-invasive) in order to achieve accurate and reliable results. In their study, Vescio et al. [20] compare classic electrocardiography with an earlobe pulse photoplethysmographic detection system. This approach seeks to evaluate the efficacy and accuracy of these techniques while keeping stress factors at bay. Their results show that earlobe measurements are strictly comparable to electrocardiogram measurements for HRV calculations. Can et al. [21] present an automatic stress detection system using smart wearable devices combining photoplethysmography and galvanic skin response measurements. Thus, they calculate stress by combining heart activity data and skin conductance data, together with data gathered from an accelerometer and a temperature sensor. One of the conclusions of their experiment is that the combined usage of the heart activity and the electrodermal activity increases the performance of the detection system. Mejía-Mejía et al. [22] note that, although HRV and pulse rate variability (PRV) are highly correlated in healthy subjects [23], there may be differences in HRV and PRV under various physiological conditions. PRV measurements are usually performed from red photoplethysmography signals acquired from different parts of the body (earlobe, finger, etc.) Their results show that the relationship between HRV and PRV is actually affected by changing physiological conditions, such as the temperature of the body. Umair et al. [24] collect and analyze the subjective opinions of users about different methods to gather HRV data. Several users participated in the experiments using different sensors in three sessions of 70 min followed by semi-structured interviews. In these interviews, subjective information is gathered on sensors' wearability, comfort, long-term use, aesthetics and social acceptance. Their results show that users prefer wrist and arm-worn devices in terms of aesthetics, wearability, and comfort, followed by chest-worn devices.

In the particular case of the usage of the HRV in a driving context, Healey and Picard [25] present a method to measure stress levels in drivers by using physiological signals. The final goal of the authors is to use this information to build adaptive systems to help drivers cope with stress, for example, by automatically diverting phone calls to a voice mail service during high-stress situations. The authors perform several tests under real driving conditions by combining several driving periods with rest periods to gather baseline measurements in low-stress situations. Drives ranged from 50 to 90 min depending on traffic conditions and were followed by a phase in which drivers had to fill out several questionnaires. Physiological sensors included electrocardiogram electrodes connected to a computer to calculate heart rates and HRV. Their results show that up to three stress levels

may be detected with an accuracy of 97.4%, with heart rates and skin conductivity metrics having the highest correlations with continuous driver stress levels.

Similarly, Jeong et al. [26] analyze a driver's cardiac activity to identify potentially dangerous situations and take prompt actions intended for accident prevention. Recognizing the complexity of measuring the physiological signals of drivers, the authors introduce a system designed to monitor the electrocardiogram (ECG) signals of drivers while they are actively operating a vehicle. This system is capable of estimating a stress index by utilizing data derived from HRV. They perform tests under real driving conditions using a designated route of approximately 200 km. To induce stress, drivers were regularly instructed to change their driving speeds. Although the authors present a first-stage pilot experiment, they conclude that the analysis of HRV is a suitable method to infer stress levels in drivers.

Munla et al. [27] analyze electrocardiogram signals to derive HRV data. They perform different analyses of HRV in the time, frequency, nonlinear and time-frequency domains to identify the most suitable parameters that vary between two levels of stress in a real driving context. Using the Stress Recognition in Automobile Driver database (DRIVEDB) dataset from PhysioBank, the authors carry out several analyses in said domains, extract the most relevant features and feed them to several classifiers. Their results indicate that stress detection may be performed with an accuracy of 83%, using a support vector machine with a radial basis function kernel (SVM-RBF). The highest prediction rates are achieved when the classifier is applied to time and non-linear parameters.

Dalmeida and Masala [28] use the aforementioned dataset to develop a predictive model to infer stress by using HRV measurements. The main goal of the authors is to develop an accurate system to monitor stress in drivers in order to reduce car crashes, as "nearly 80% of road incidents are due to drivers being under stress". The predictive model is obtained by computing different supervised machine learning models including the K-Nearest Neighbor (KNN), Support Vector Machines (SVM), Multilayer Perceptron (MLP), Random Forest (RF) and Gradient Boosting (GB). Also, they perform experiments with real users by deriving stress levels from HRV measurements obtained from a smartwatch device. The authors conclude that HRV features are good markers for stress detection and that the best results are obtained with the MLP model, with sensitivity scores of 80%.

Apart from stress, personality is also something that may influence drivers' behavior. In fact, it has already been demonstrated that attitudes and personality traits are general predictors of behavior [29]. Nevertheless, they are poor predictors of specific actions [29] because of the complexities involved. Thus, previous work has tried to determine if personality traits affect specifically the behavior of drivers or not. For example, by analyzing personality and mood, Matthews et al. [30] try to determine the predisposition to driver stress. The authors perform four studies in which they try to find out how personality correlates with driver stress by using a driver behavior (DBI) questionnaire. These studies include an analysis of the relations between the stress of drivers and different dimensions of personality (extroversion, neuroticism and psychoticism); of the role of intra-personal and situational frustration and annoyance in driver stress; of the relationships between driver stress and self-reported attentional efficiency and of the DBI as a predictor of emotional state. The authors conclude that, despite some limitations, self-reported driver stress levels correlate with various measures of personality, cognitive appraisal and emotion.

Iversen and Rundmo [31] present a study based on a questionnaire survey performed by a set of random drivers selected from the Norwegian driver's license register. They aim to analyze a set of personality characteristics of drivers and their relationship with risky driving and accidents. These characteristics include locus of control, driver anger, sensation seeking and normlessness. Their results show that there is a tendency for sensation seekers and drivers with high normlessness and anger scores to report more speeding and rule infringements. They also report that these particular characteristics influence the involvement in traffic accidents. In the study conducted by Zhao et al. [32], an investigation was undertaken to assess the stress levels experienced by individuals of varying age groups,

specifically older and younger men, while driving along straight road segments and navigating through intersections. The researchers collected data from experimental journeys spanning a distance of 22.4 km, monitoring stress levels via the CAN-Bus (Controller Area Network) and gathering self-reported Stress (SRS) data in conjunction with physiological measurements. The findings of this study indicated that older drivers self-reported experiencing less stress than their younger counterparts. However, it was noteworthy that these older drivers were inclined to underestimate their perceived stress levels, which became evident upon scrutinizing the physiological parameters.

Moreover, Machin and Sankey [33] carry out a study in which they try to analyze the relationship between personality factors, risk perceptions and driving behavior in young drivers. The authors perform an online survey among 159 Australian students aged between 17 and 20, using personality variables found in previous work to predict driving behavior. These variables assess anxiety, anger, excitement-seeking, altruism, and normlessness. Risk perception variables include worry and concern, the likelihood of an accident, efficacy and aversion to risk-taking, and, finally, driving behavior that was limited to speeding. The authors conclude that speeding is strongly related to a lower dislike of risk-taking. Also, a dislike of risk-taking acts as a mediator of the influence of other aspects of personality on speeding.

More recently, Lucidi et al. [34] also analyze the role of personality characteristics and attitudes toward traffic safety while considering a possible moderating role played by age. Their study analyzes the data gathered with a questionnaire filled by 1286 drivers from three different age groups. This questionnaire included elements to assess personality characteristics (anxiety, hostility, excitement seeking, altruism and normlessness), positive attitudes toward traffic safety, risky driving behaviors (errors, lapses and traffic violations), accident involvement and traffic fines. The authors conclude that anxiety and hostility predicted attitudes toward traffic safety only in older drivers, whereas altruism affected positive attitudes toward safety significantly and positively only in young and middle-aged drivers. Also, positive attitudes toward traffic safety were globally related to fewer violations and errors in all the groups. Moreover, their results show that excitement-seeking impacts errors and lapses only in young drivers and only errors in middle-aged drivers.

Zicat et al. [35] explored the relationship between driving, attitudes, personality and cognition. The authors performed several experiments with 152 young participants using the STISIM (https://stisimdrive.com/ (accessed on 18 December 2023)) driving simulator and measuring speeding and lane deviations. In the evaluation, attitudes consist of a score towards road safety; personality is assessed with scores on anxiety, anger and sensation seeking and cognition is analyzed with four tests measuring several cognitive functions (attention, processing speed, executive function, visuospatial perception, memory, and psychomotor performance). The authors conclude that there is a significant relationship between attitudes, personality characteristics, and speeding in young drivers. This confirms the conclusions drawn in previous work, which were mainly derived from questionnaires. On the contrary, the relationship predicting lane deviations from driving attitudes and personality was found to be not significant.

In a similar fashion, Linkov et al. [36] undertake an examination of the connection between personality traits and driving behavior. This investigation involved the use of a truck-driving simulator. The assessment of driving performance entails the measurement of parameters such as speed and lateral deviations. A total of 41 professional drivers take part in a series of experiments, each involving various driving scenarios. Their driving performance was compared with their personality characteristics (neuroticism, extroversion, openness to experience, agreeableness and conscientiousness), sensation seeking, and present-time perspective. The authors also conclude that a distinct correlation exists between personality variables and the manifestation of safe driving behavior.

Landay et al. [37] explored the connections between various aspects of personality and driving accidents, focusing mainly on professional truck drivers. In contrast to prior research, this study employs company records of accidents as its data source instead of

relying on self-reported accident information. Moreover, the authors utilize driver self-ratings on the Hogan Personality Inventory (HPI), with a notable difference being the use of the 41 lower-order facets instead of the seven personality facets that the HPI may yield. The findings indicate that these broader personality facets, for the most part, exhibit relations with accidents, consistent with previous research. However, not all the lower-order facets of HPI are individually associated with accidents. Consequently, the authors suggest that only certain lower-order facets may account for the association with accidents. Thus, a more nuanced view of personality is needed in the analyses. They conclude that there is a broad range of concrete associations that can be found between accident involvement and personality.

Table 1 shows a summary of the different approaches followed in previous work.

A review of previous studies indicates a lack of substantial influence of personality on driving behavior and cultural distinctions. However, our study revealed a subtle difference between Spain and Germany. This distinction was not only evident in the overall overview but also observable in the varying personality traits. These findings underscore the significance of incorporating cultural aspects and personality traits into the development of systems that predict stress and driving behavior.

**Table 1.** The approach followed in previous work.

| Approach | References |
| --- | --- |
| Assessment of stress based on the HRV | [16–19,21,25–28] |
| Biomarkers and sensor technologies to assess the HRV | [16,20–22,24,27,28] |
| Usage of the HRV in a driving context | [25–28] |
| Personality traits and behaviour of drivers | [30,31,33–37] |
| Self-reported stress while driving | [32] |

## 3. Method

This section introduces the methodology, metrics, and experimental setup, followed by a detailed description of the experimental procedure.

### 3.1. Participants

To investigate the impact of stress on road safety and compare the influences of various personality traits, two groups of drivers were selected from Germany and Spain. Each participant was equipped with a Polar H10 chest strap, which collected physiological data. The primary focus here was on gathering data regarding the time intervals between consecutive heartbeats, also known as RR intervals. Before commencing their driving sessions, participants were required to complete a Revised Eysenck Personality Questionnaire (EPQR-S). Following the questionnaire, participants underwent a relaxation phase during which they listened to a melody for 5 min while their RR intervals were collected. This phase was designed to establish a baseline level of relaxation. Subsequently, participants continued with the driving phase, which was 25 min long. Throughout this period, both the RR intervals and driving behavior data were measured and recorded. Upon completing the driving phase, participants were again asked to answer the EPQR-S questionnaire. The study aimed to separate a distinct connection between the drivers' personality traits and the stress level experienced while driving. Based on their responses to the EPQR-S questionnaire, participants were categorized into one of the four selected personality types. The outcomes of this research would facilitate a comparative analysis of the relationship between distinct personality traits and the experienced stress during driving.

A total of 71 drivers took part in the study, with 38 from Germany and 33 from Spain. The mean age of the participants was $25.84 \pm 4.83$ years for German drivers and $25.27 \pm 3.81$ years for Spanish drivers. All participants held valid driving licenses. None of the participants were professional drivers, defined as individuals driving trucks, taxis, buses, or sports cars. The participants were categorized as amateur drivers. All participants were residing in Germany or Spain. They volunteered for the study and did not receive any

payment or benefits for their participation. The selection criteria included being in good health, having no history of heart-related diseases that could affect HRV, and possessing normal vision. Additionally, participants were explicitly instructed not to consume drugs, alcohol, or any other beverages or medications before the experiment.

### 3.2. Heart Rate Variability (HRV)

HRV can be defined as the fluctuation in the time intervals between adjacent heartbeats. It has a strong relationship with the activity of the autonomic nervous system and is employed in order to detect heart problems and as a stress marker. In this study we use the following methods to quantify the HRV:

- Average Heart rate (bpm): It is the average number of beats per minute during the test. It is a measure in the time domain within milliseconds.
- Average RR interval (ms): It is the average time between successive heartbeats during the test. It is a measure in the time domain.
- LF/HF ratio: It is calculated by dividing the low-frequency (LF) power (0.04–0.15 Hz) modulated by the sympathetic and parasympathetic nervous system by the high-frequency (HF) power (0.15–0.4 Hz) associated with the parasympathetic nerve activity. It is a measure in the frequency domain.

The heart signal is monitored using a H10 sensor and Pro strap from Polar Electro, Professorintie 5, 90440 Kempele, Finland, which were purchased from their web shop. See in Figure 1.The precision of this device is similar to that obtained by a Holter ECG [38], although unlike this it is not intrusive. The driver can wear it while driving without multiple cables interfering with vehicle handling. In addition, its cost is low.

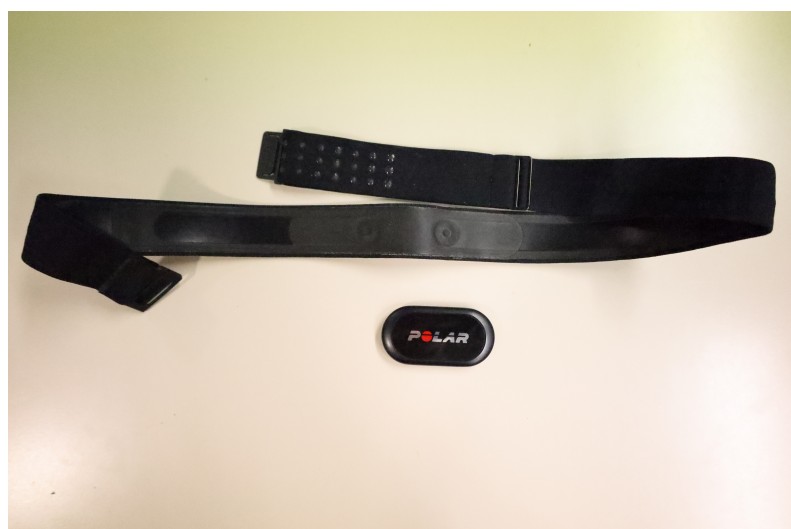

**Figure 1.** Polar H10 chest strap.

The heart signal is prone to noise and artifacts. This is due to the involuntary movement of the subject and factors such as sensor type, electromagnetic noise, connections, etc. Therefore, it is necessary to correct or eliminate these artifacts to avoid incorrect readings. In this work, we employ an artifact correction method based on the comparison of each RR interval against a local average interval. The local average is calculated by applying median filtering on the RR time series. This prevents the local average from being influenced by single outliers in the RR time series.

HRV analysis and signal preprocessing are performed using Kubios version 3.5. Kubios [39] is an analysis software that is used by researchers to evaluate cardiovascular health and the effects of stress. It has been scientifically validated and has more than 1000 citations.

### 3.3. Personality Survey

The Eysenck Personality Questionnaire (EPQR) is one of the most widely used personality trait questionnaires in the world. It consists of 100 items from which the dimensions of extroversion, neuroticism, and psychoticism are evaluated [40].

A high score implies sociability, carefreeness, spontaneity, and liveliness in the extroversion dimension. On the contrary, a low score indicates introversion and avoidance of excessive social contact. This dimension is linked to the ascending reticular activation system. It is the social dimension. In the dimension of neuroticism, a high score involves anxiety, depression, low self-esteem, emotivity and tension. On the contrary, a low score is related to emotional stability. This is the emotional dimension. In the dimension of psychoticism, a high score is related to aggressiveness, egocentrism, impersonalization and the absence of empathy. This dimension correlates well with psychopathy and crime. According to Eysenck, it is the impulsive dimension. All these dimensions are not mutually exclusive, and the personality traits can be mixed.

The primary limitation associated with the Eysenck Personality Questionnaire (EPQR) lies in its length, which can be impractical for research studies involving an extensive assessment of variables. To address this issue, a condensed version of the questionnaire, known as the Eysenck Personality Questionnaire Revised-Abbreviated (EPQR-A), was developed by Francis et al. [41]. The EPQR-A includes 24 items, with six items for each of its subscales. The conception of the EPQR-A involved a sample of 685 undergraduate students across England, Canada, and Australia. The selection of questionnaire items for this abbreviated version was based on an evaluation of item-total correlations for each of the subscales. In our study, we utilized a questionnaire derived from the EPQR questionnaire.

### 3.4. Driving Simulator

The driving assessment was conducted using the City Car Driving—Car Driving Simulator software V1.5, from Forward Development, https://citycardriving.com/ (accessed on 18 December 2022), which is known for its implementation of advanced car physics that provides a high level of realism. The City Car Driving simulator software V1.5 was purchased on their online store. Furthermore, the simulator allows for the customization of pedestrian and road user behaviors, as well as the adjustment of traffic density to replicate real-world driving conditions and repeatable scenarios. One of the main advantages of the simulation is that everything occurs in a safe environment where errors and mistakes have no real-world consequences. The simulator's immersive setup included three monitors and a car seat, as shown in Figure 2.

For the experiment, the following scenario was configured:

- Traffic density was set to normal.
- Various challenging events were enabled, including drivers traveling on the opposite lane, abrupt lane changes just in front of the driver's car, pedestrians crossing the road in unauthorized locations, malfunctioning traffic lights, and more.
- The scenario involved a combination of narrow streets with unsupervised pedestrian crossings and multi-lane roads, including highways.

This was designed to amplify the number of stressors and stressful events occurring during the experiment. Additionally, the driving simulator adheres to European traffic rules and issues warnings when users fail to comply with any rule, such as speeding, not using turn signals, or not stopping at pedestrian crossings. In cases where the driver commits an infraction or is involved in a traffic accident, the event is recorded and saved in a log file.

For input during the simulations, we utilized a Logitech G29, an electronic steering wheel explicitly designed for driving video games. This device features realistic force feedback and includes a set of three pedals along with a gearbox, collectively contributing to an immersive experience within the virtual environment.

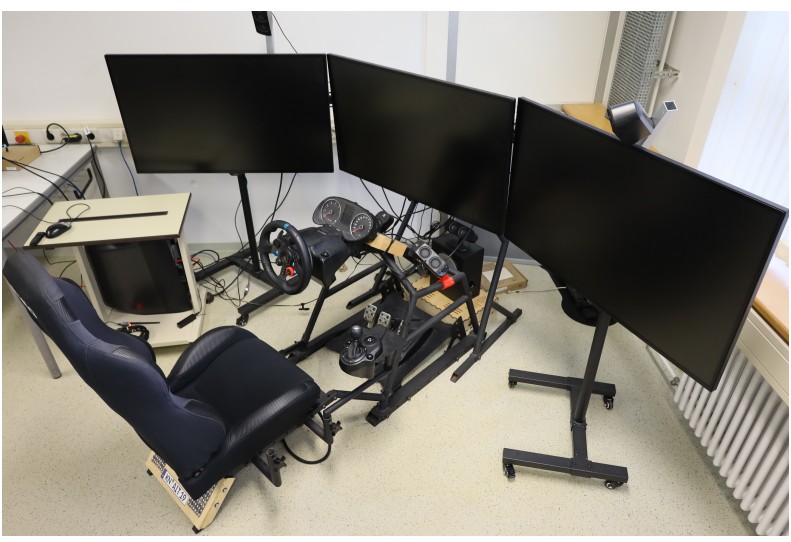

**Figure 2.** Driving simulator hardware.

*3.5. Procedure*

The study was conducted following these steps as illustrated in Figure 3.

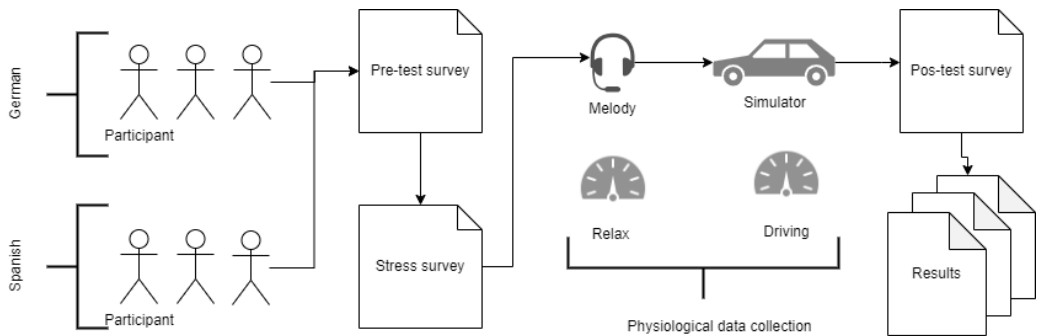

**Figure 3.** Experiment design.

1.  The participants were equipped with the Polar H10 chest strap and completed an initial survey. The primary purpose of this survey was to gather information regarding the driver's characteristics (age, gender, driving experience) and their current physical and mental state.
2.  Each participant was provided with headphones and listened to Mozart's "Sonata for Two Pianos in D major". This musical composition was chosen due to its well-documented usefulness in inducing relaxation, and supporting a peaceful mood. Previous work has already studied music as a good method to manage stress [42]. Furthermore, numerous studies have demonstrated the benefits of music in improving short-term memory and concentration.
3.  During the driving phase, participants engaged in a 25-min drive. In this driving test, we closely monitored the participant's heart signal, driving behavior, and obedience to traffic regulations. The driving routes had a consistent length of 5 km, and their difficulty level was intentionally comparable by maintaining consistent levels of vehicular and pedestrian traffic throughout. At the start of each route, the driving simulator assigned participants a set number of points. Points were subsequently deducted each time the participant committed a traffic infraction. When the score reached zero, the participant was required to repeat the route. This approach aimed to keep the participants fully engaged in the driving task, simulating the focus expected in a real-world driving environment. The 25-min duration of the driving phase was chosen to ensure that the stress data collected would be sufficiently valid and representative.

4. The participants filled out the EPQR-S questionnaire to assess the dimensions of extroversion, neuroticism, and psychoticism.

## 4. Results

The following sections present the results of HRVs in relation to the analyzed personality traits within the dimensions of neuroticism and extroversion, respectively. Additionally, the results from Germany and Spain are presented.

### 4.1. Neuroticism Dimension in Germany

Table 2 describes the groups built according to the neuroticism score. Group A is made up of drivers who scored high neuroticism and Group B of those who obtained a lower score. Table 3 shows the average value of the HRV measurements obtained by the two groups of drivers. High values of heart rate and LF/HF ratio, and low values of RR intervals are associated with high-stress levels.

In the relaxation phase, Group A obtained higher results in all the measurement values that indicate more stress than Group B, although the differences are not significant. The average heart rate in Group A is 6% higher than in Group B, $t(36) = 1.275$, $p > 0.05$. In the RR intervals, the average value obtained by Group A is 7% lower than in Group B, $t(36) = -1.443$, $p > 0.05$. Finally, the average value of the LF/HF ratio in Group A is twice that obtained by Group B, $t(9.887) = 1.682$, $p > 0.05$.

During the driving phase, the Group A drivers were the ones who presented worse indicators, although there are only significant differences in the LF/HF ratio. In the average heart rate, Group A drivers obtained a value 4% higher than Group B drivers, $t(36) = 0.855$, $p > 0.05$. In the RR interval, Group A obtained a 4% lower value than Group B, $t(36) = -1.002$, $p > 0.05$. Finally, in the LF/HF ratio, the drivers of Group A doubled the value obtained by Group B, $t(36) = -3.157$, $p < 0.05$.

Figure 4 captures the difference between the driving and relaxation phases for each of the HRV measures. The LF/HF ratio is the only measure in which we found significant differences between both groups, $t(36) = -3.157$, $p < 0.05$. Drivers with a low neuroticism score increased this variable twice as much as drivers with a higher neuroticism score.

**Table 2.** German drivers grouped by the neuroticism score.

| Group | Number of Drivers | Neuroticism Thresholds | Average Neuroticism Score |
|---|---|---|---|
| A | 10 | [4–6] | $5.00 \pm 0.82$ |
| B | 28 | [0–3] | $1.50 \pm 1.00$ |

**Table 3.** HRV according to the neuroticism score obtained by German drivers.

| Measurement | Phase | Group | Average Value |
|---|---|---|---|
| Heart Rate | Relaxation | A | 78.4 bpm $\pm$ 6.6 bpm |
| | | B | 74.0 bpm $\pm$ 10.2 bpm |
| | Driving | A | 82.4 bpm $\pm$ 6.4 bpm |
| | | B | 79.4 bpm $\pm$ 9.8 bpm |
| RR Intervals | Relaxation | A | 770.9 ms $\pm$ 67.5 ms |
| | | B | 825.3 ms $\pm$ 111.6 ms |
| | Driving | A | 732.3 ms $\pm$ 59.4 ms |
| | | B | 765.3 ms $\pm$ 96.6 ms |
| LF/HF ratio | Relaxation | A | $3.7 \pm 3.4$ |
| | | B | $1.9 \pm 1.2$ |
| | Driving | A | $6.2 \pm 4.4$ |
| | | B | $3.1 \pm 1.6$ |

The HRV of German drivers, according to their neuroticism score, is divided into two phases: relaxation and driving. Each phase is further divided into two subgroups, [A] and [B], and grouped in the table by grey or white blocks.

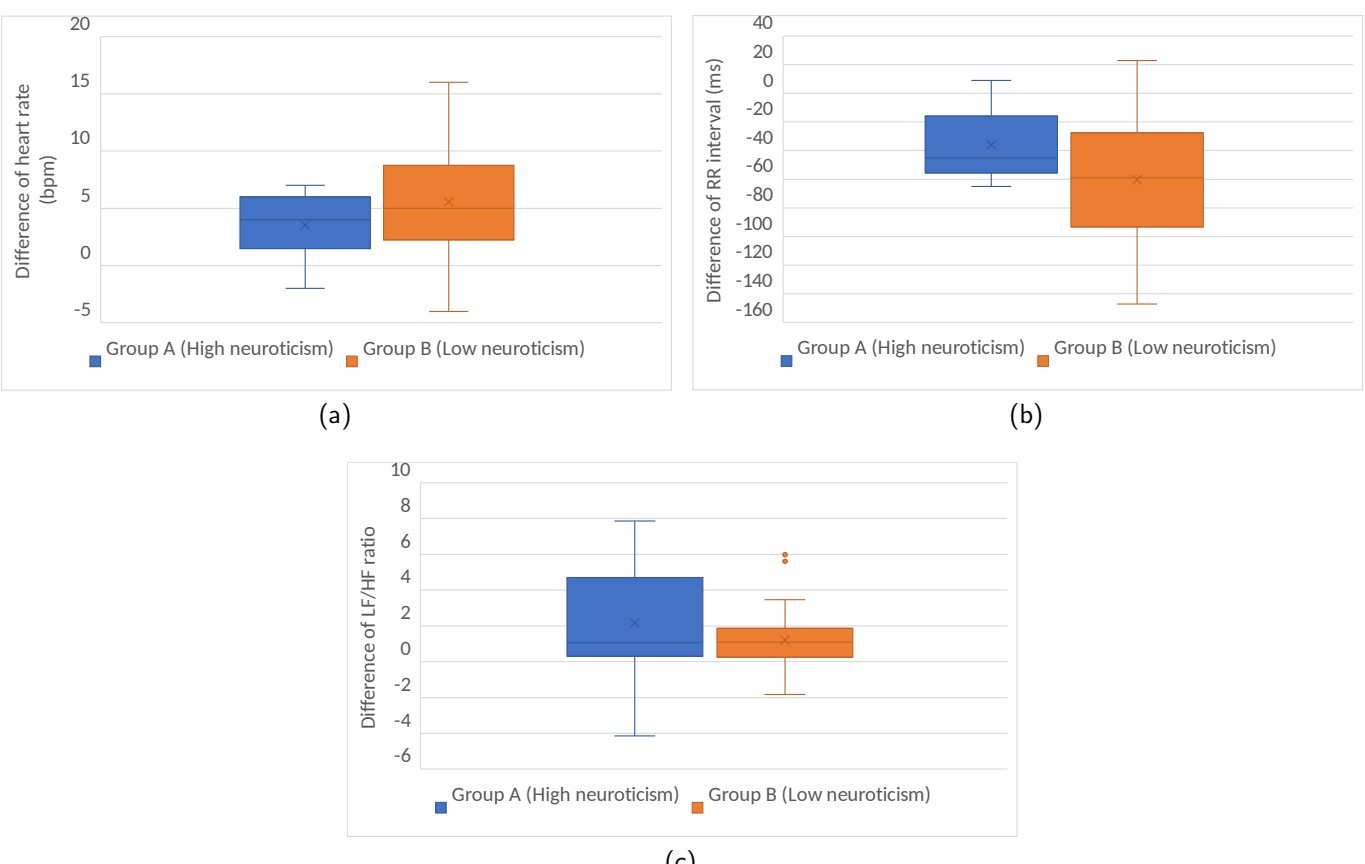

**Figure 4.** Difference of HRV between the driving and relaxation phases according to the neuroticism level in Germany. (**a**) Heart Rate; (**b**) RR Interval; (**c**) LF/HF ratio.

### *4.2. Extroversion Dimension in Germany*

Table 4 describes the groups that have been established considering the score obtained by the participants in the extroversion dimension. Group A is clustered by drivers who scored high in extroversion and Group B by drivers who scored high in introversion. Table 5 shows the results of the HRV.

In the relaxation phase, Group A was the one that obtained values more associated with suffering stress than Group B. In the case of heart rate, the average value of Group A was 3% higher than that of Group B, t(36) = 0.762, $p > 0.05$. In the LF/HF ratio, the percentage was 9%, t(36) = 0.304, $p > 0.05$. Regarding the RR, Group A obtained an average value 2% lower than Group B, t(36) = $-0.557$, $p > 0.05$. However, there are no significant differences between the groups for any of the three measures.

During the driving phase, the opposite happens and the drivers with a high score in introversion (Group B) were the ones who presented worse average values in the stress indicators, although it is important to highlight that the differences between both groups are not significant. The average heart rate was 0.4% higher than the value obtained by Group A, the average RR was 1% lower, and the LF/HF ratio was 1% higher. Figure 5 captures the difference between the driving and the relaxation phase in HRV. In both groups, the driving task increased the stress level in a similar manner, the *t*-test indicates that there are no significant differences.

**Table 4.** German drivers grouped by extroversion score.

| Group | Number of Drivers | Extroversion Thresholds | Average Extroversion Score |
|-------|-------------------|-------------------------|----------------------------|
| A | 23 | [4–6] | 4.82 ± 0.82 |
| B | 15 | [0–3] | 1.60 ± 1.05 |

**Table 5.** HRV according to the extroversion score obtained by German drivers.

| Measurement | Phase | Group | Average Value |
|-------------|-------|-------|---------------|
| Heart Rate | Relaxation | A | 76.1 bpm ± 10.4 bpm |
| | | B | 73.7 bpm ± 8.1 bpm |
| | Driving | A | 80.2 bpm ± 10.2 bpm |
| | | B | 80.5 bpm ± 7.4 bpm |
| RR Intervals | Relaxation | A | 803.3 ms ± 116.4 ms |
| | | B | 822.7 ms ± 83.4 ms |
| | Driving | A | 759.4 ms ± 102.0 ms |
| | | B | 751.9 ms ± 66.7 ms |
| LF/HF ratio | Relaxation | A | 2.5 ± 2.5 |
| | | B | 2.3 ± 1.4 |
| | Driving | A | 3.9 ± 3.4 |
| | | B | 3.9 ± 2.0 |

The HRV of German drivers, according to their extroversion score, is divided into two phases: relaxation and driving. Each phase is further divided into two subgroups, [A] and [B], and grouped in the table by grey or white blocks.

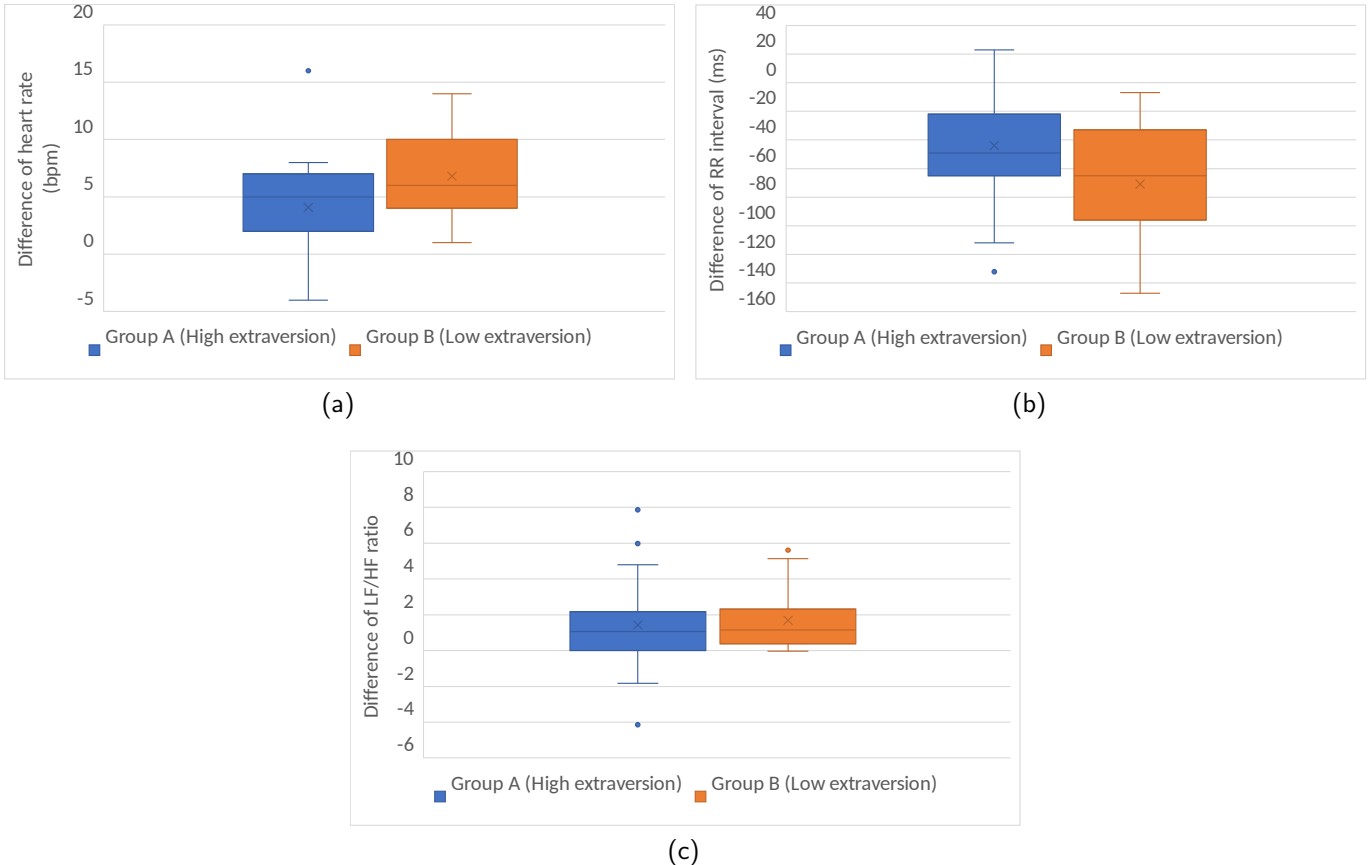

(a)

(b)

(c)

**Figure 5.** Difference of HRV between the driving and relaxation phase according to the extroversion level in Germany. (**a**) Heart Rate; (**b**) RR Interval; (**c**) LF/HF ratio.

*4.3. Psychoticism Dimension in Germany*

Table 6 describes the groups that have been composed taking into account the score obtained by the participants in the psychoticism dimension. Group A is made up of drivers who obtained a score equal to or greater than 2 in this dimension. Group B is built by drivers who scored less than 2. A high value in this dimension is correlated with aggression, impulsivity, aloofness, and antisocial behavior. Table 7 shows the results of the HRV analysis.

In the relaxation phase, the drivers with the lowest score in this dimension (Group B) present worse indicators from the point of view of stress compared to the participants with the highest scores (Group A). However, the differences are not significant according to the *t*-test. In the case of heart rate, the average value of Group B is 7% higher than the average value of Group A, t(36) = −1.702, $p > 0.05$. In the LF ratio/HF ratio, the percentage is 69%, t(36) = 1.554, $p > 0.05$. Considering the RR, Group B obtained an average value 7% lower than Group A, t(36) = −1.702, $p > 0.05$.

In the driving phase, drivers from Group B were also those who suffered greater stress according to the three stress measurements, although the differences between both groups are not significant either. In the average heart rate, Group B drivers obtained an average value 4% higher than Group A, t(36) = −1.190, $p > 0.05$. In the RR interval, Group B obtained a value 4% lower than Group A, t(36) = 1.148, $p > 0.05$. Finally, the average LF/HF ratio from Group B was 34 % higher than Group A, t(36) = −1.180, $p > 0.05$.

Figure 6 captures the difference between the driving and relaxation phases in the stress indicators. All the participants experienced an increase of the stress level during the driving task and this was similar in the two groups of drivers analyzed as there are no significant differences according to the *t*-test.

**Table 6.** German drivers grouped by psychoticism score.

| Group | Number of Drivers | Psychoticism Thresholds | Average Psychoticism Score |
|:-----:|:-----------------:|:-----------------------:|:--------------------------:|
| A | 16 | [2–6] | 2.37 ± 0.50 |
| B | 22 | [0–1] | 0.64 ± 0.49 |

**Table 7.** HRV according to the psychoticism score obtained by German drivers.

| Measurement | Phase | Group | Average Value |
|:-----------:|:-----:|:-----:|:-------------:|
| Heart Rate | Relaxation | A | 72.1 bpm ± 7.4 bpm |
| | | B | 77.3 bpm ± 10.4 bpm |
| | Driving | A | 78.3 bpm ± 8.7 bpm |
| | | B | 81.8 bpm ± 9.3 bpm |
| RR Intervals | Relaxation | A | 841.1 ms ± 91.7 ms |
| | | B | 789.1 ms ± 108.7 ms |
| | Driving | A | 775.8 ms ± 85.9 ms |
| | | B | 742.4 ms ± 90.2 ms |
| LF/HF ratio | Relaxation | A | 1.7 ± 1.2 |
| | | B | 2.9 ± 2.5 |
| | Driving | A | 3.3 ± 1.8 |
| | | B | 4.4 ± 3.5 |

The HRV of German drivers, according to their psychoticism score, is divided into two phases: relaxation and driving. Each phase is further divided into two subgroups, [A] and [B], and grouped in the table by grey or white blocks.

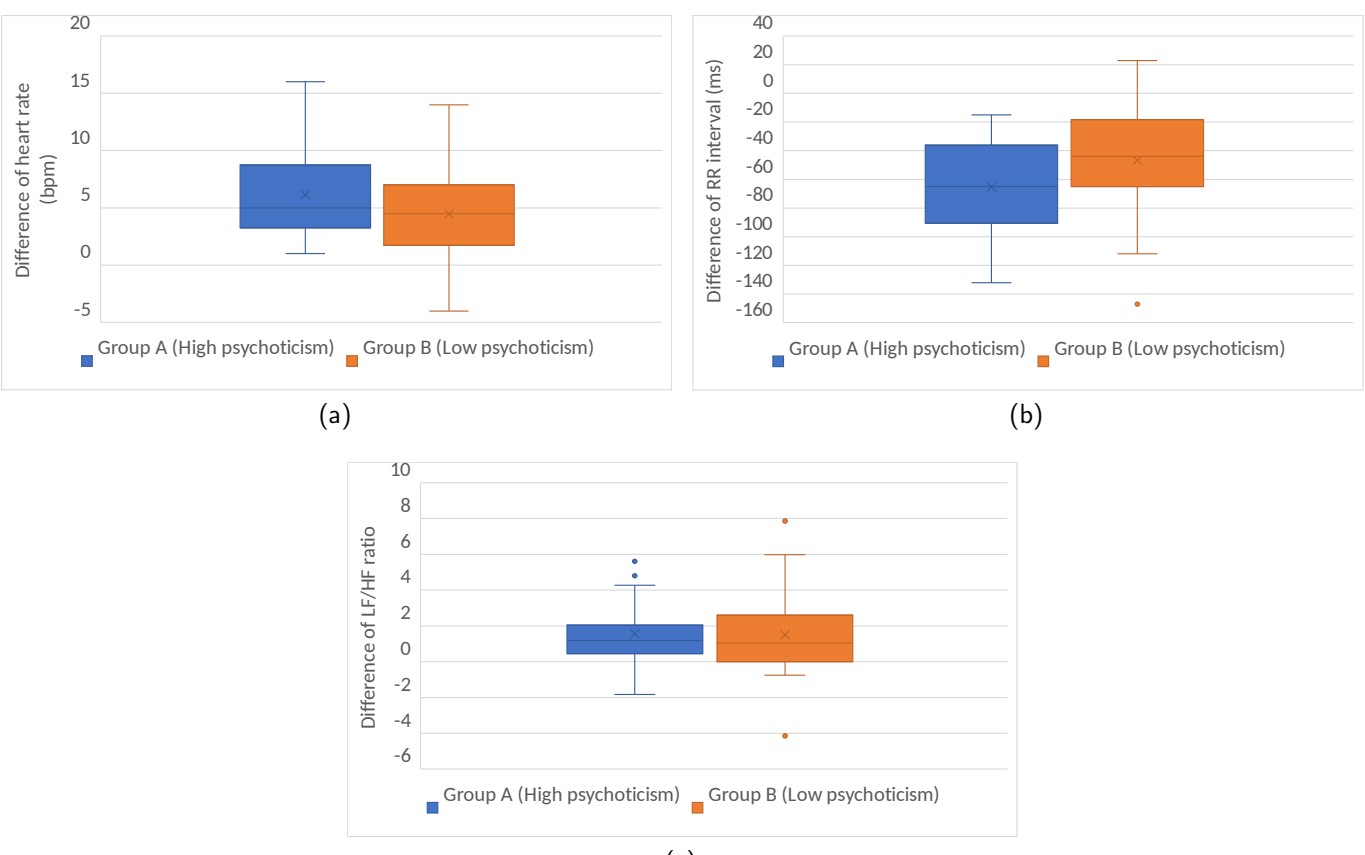

**Figure 6.** Difference of HRV between the driving and relaxation phase according to the psychoticism level in Germany. (**a**) Heart Rate; (**b**) RR Interval; (**c**) LF/HF ratio.

### 4.4. Neuroticism Dimension in Spain

Table 8 describes the groups established taking into account the neuroticism score. Group A (high neuroticism) is made up of drivers with scores equal to or greater than 4. Group B (low neuroticism) is made up of drivers with scores lower than 4. Table 9 shows the results of the HRV analysis.

In the relaxation phase, we observed significant differences in heart rate and RR interval. The average heart rate of Group A is 32% higher than that of Group B, t(31) = 5.080, $p < 0.05$. The RR interval of Group A is 25% lower than that of Group B, t(31) = −4.154, $p < 0.05$. In the case of the LF/HF ratio, the value of Group A is 73% higher than that of Group B, but the difference is not significant, t(31) = 2.030, $p > 0.05$.

In the driving phase, the group made up of the drivers who obtained a higher score in neuroticism (Group A) are also those who present worse stress indicators. In addition, in this case, there are significant differences between the three measures. The average heart rate of Group A is 29% higher than that of Group B, t(31) = 5.139, $p < 0.05$. The RR interval of Group A is 24% lower than Group B, t(31) = −4.061, $p < 0.05$. Finally, the LF/HF ratio in Group A is 90% lower than in Group B, t(31) = 2.982, $p < 0.05$.

Analyzing the difference between the driving phase and the relaxation phase Figure 7, we can see that the driving test has a similar impact on the two groups of drivers since the differences between the two groups are not significant in any of the three stress measurements (heart rate, RR interval, and LF/HF ratio) according to the *t*-test.

**Table 8.** Spanish drivers grouped by neuroticism score.

| Group | Number of Drivers | Neuroticism Thresholds | Average Neuroticism Score |
|---|---|---|---|
| A | 9 | [4–6] | 5.33 ± 0.87 |
| B | 24 | [0–3] | 1.04 ± 0.75 |

**Table 9.** HRV according to the neuroticism score obtained by Spanish drivers.

| Measurement | Phase | Group | Average Value |
|---|---|---|---|
| Heart Rate | Relaxation | A | 91.6 bpm ± 12.7 bpm |
| | | B | 69.3 bpm ± 10.7 bpm |
| | Driving | A | 94.8 bpm ± 8.6 bpm |
| | | B | 73.7 bpm ± 11.1 bpm |
| RR Intervals | Relaxation | A | 667.0 ms ± 108.4 ms |
| | | B | 887.8 ms ± 144.3 ms |
| | Driving | A | 637.0 ms ± 59.9 ms |
| | | B | 833.6 ms ± 139.4 ms |
| LF/HF ratio | Relaxation | A | 5.1 ± 3.8 |
| | | B | 2.9 ± 2.2 |
| | Driving | A | 6.9 ± 3.0 |
| | | B | 3.6 ± 2.7 |

The HRV of Spanish drivers, according to their neuroticism score, is divided into two phases: relaxation and driving. Each phase is further divided into two subgroups, [A] and [B], and grouped in the table by grey or white blocks.

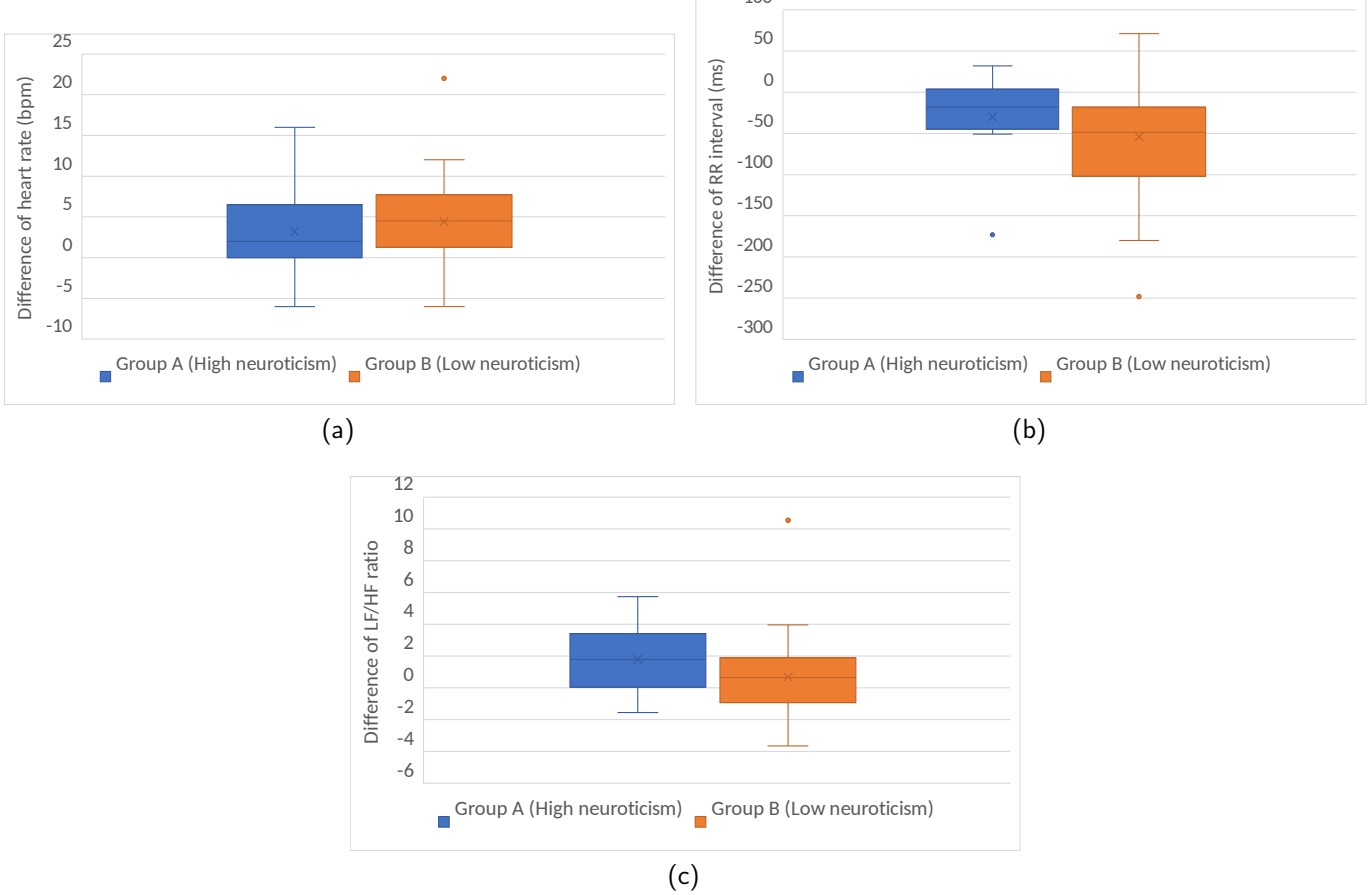

**Figure 7.** Difference of HRV between the driving and relaxation phase according to the neuroticism level in Spain. (**a**) Heart Rate; (**b**) RR Interval; (**c**) LF/HF ratio.

*4.5. Extroversion Dimension in Spain*

Table 10 details the groups created taking into account the extroversion score. Group A is made up of drivers with scores equal to or greater than 4. Group B is made up of drivers with scores lower than 4. Table 11 shows the results of the HRV analysis.

In the relaxation phase, extroverted drivers (Group A) show higher stress indicators than introverted drivers (Group B), but the difference is only significant in the LF/HF ratio. The average heart rate of Group A is 6% higher than that of Group B, t(31) = 0.842, $p > 0.05$. The RR interval of Group A is 4% lower than that of Group B, t(31) = −0.541, $p > 0.05$. In the case of the LF/HF ratio, the value of Group A is two times that of Group B, t(31) = 2.0342, $p < 0.05$.

During the driving phase, the drivers with a low score in extroversion (Group B) are the ones who present worse average values in the stress indicators, although it is important to note that the differences between both groups are not significant. The average heart rate of Group B is 2% higher than that of Group A, t(31) = −0.251, $p > 0.05$. The RR interval of Group B is 4% lower than that of Group A, t(31) = 0.621, $p > 0.05$. Finally, the average LF/HF ratio is the same in both groups, t(31) = −0.011, $p > 0.05$.

Figure 8 captures the difference between the driving phase and the relaxation phase in the HRV. It can be seen that the driving task affects introverted drivers (Group B) more than extroverted drivers (Group A). There are significant differences in the three measures. During driving, the average heart rate in Group B increases six times more than Group A, t(31) = −3.008, $p < 0.05$. The LF/HR ratio grows two times more, t(31) = 2.927, $p < 0.05$. The RR interval decreases four times, t(31) = −2.440, $p < 0.05$.

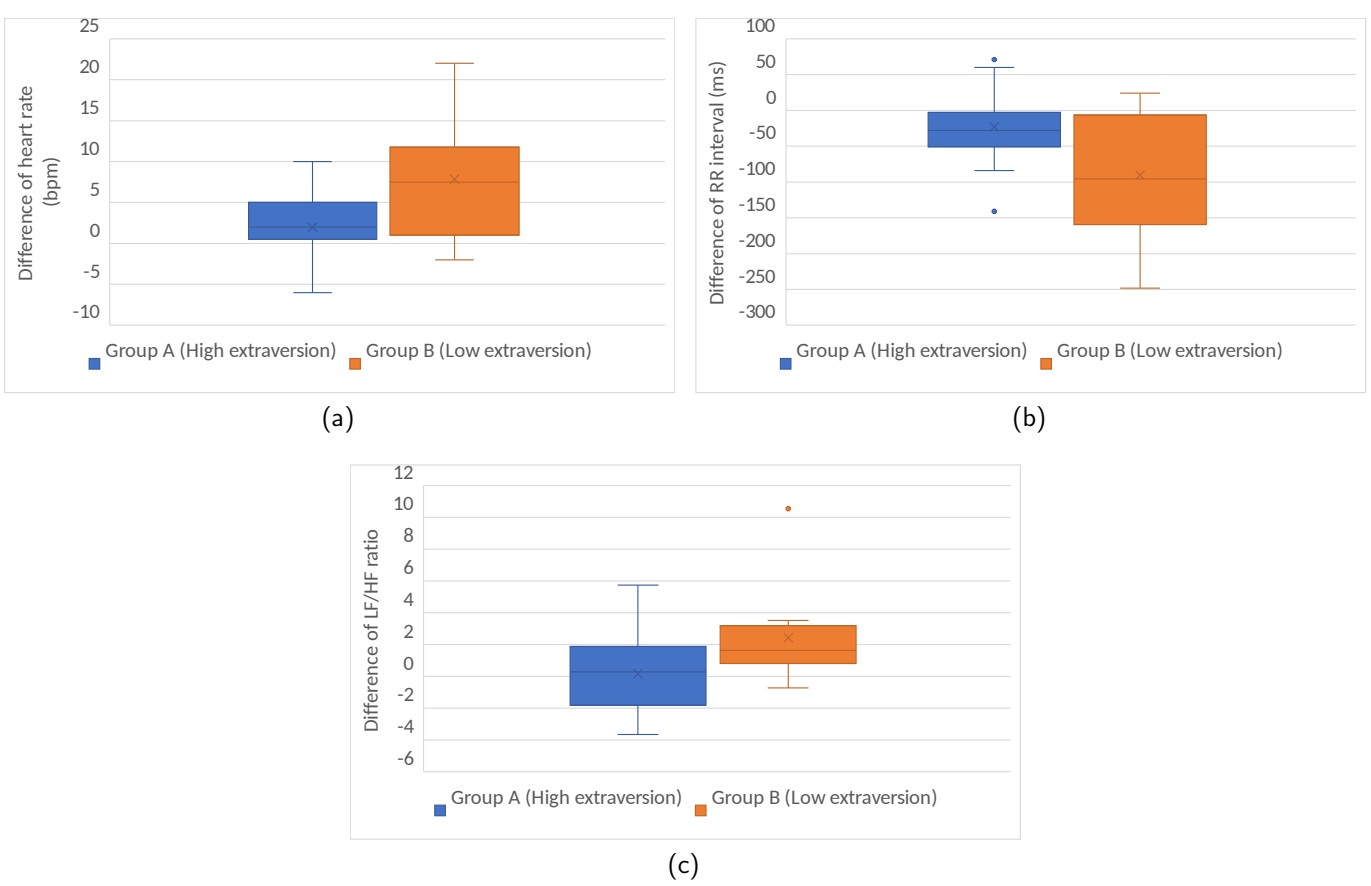

**Figure 8.** Difference of HRV between the driving and relaxation phase according to the extroversion level in Spain. (**a**) Heart Rate; (**b**) RR Interval; (**c**) LF/HF ratio.

**Table 10.** Spanish drivers grouped by extroversion score.

| Group | Number of Drivers | Extroversion Thresholds | Average Extroversion Score |
|---|---|---|---|
| A | 21 | [4–6] | 5.28 ± 0.34 |
| B | 12 | [0–3] | 0.92 ± 0.17 |

**Table 11.** HRV according to the extroversion score obtained by Spanish drivers.

| Measurement | Phase | Group | Average Value |
|---|---|---|---|
| Heart Rate | Relaxation | A | 77.0 bpm ± 16.5 bpm |
| | | B | 72.4 bpm ± 12.0 bpm |
| | Driving | A | 79.0 bpm ± 16.0 bpm |
| | | B | 80.3 bpm ± 10.4 bpm |
| RR Intervals | Relaxation | A | 815.5 ms ± 180.9 ms |
| | | B | 848.6 ms ± 144.3 ms |
| | Driving | A | 792.4 ms ± 172.9 ms |
| | | B | 758.2 ms ± 105.0 ms |
| LF/HF ratio | Relaxation | A | 4.3 ± 3.1 |
| | | B | 2.1 ± 1.6 |
| | Driving | A | 4.5 ± 3.0 |
| | | B | 4.5 ± 3.4 |

The HRV of Spanish drivers, according to their extroversion score, is divided into two phases: relaxation and driving. Each phase is further divided into two subgroups, [A] and [B], and grouped in the table by grey or white blocks.

### 4.6. Psychoticism Dimension in Spain

Table 12 presents the groups built taking into account the psychoticism score. Group A is made up of drivers with scores equal to or greater than 2. Group B is made up of drivers with scores lower than 2. Table 13 shows the results of the HRV analysis.

In the relaxation phase, drivers from Group A show higher average stress indicators than drivers from Group B. However, the differences between them are not significant. The average heart rate of Group A is 8% higher than that of Group B, $t(31) = 0.956$, $p > 0.05$. The RR interval of Group A is 8% lower than that of Group B, $t(31) = -1.010$, $p > 0.05$. In the case of the LF/HF ratio, the value of Group A is 68% higher than that of Group B, $t(6.638) = 1.155$, $p > 0.05$.

During the driving phase, the situation is similar, and the drivers with high scores in psychoticism are the ones who have worse average values in stress measurements. However, the differences between both groups are also not significant. The average heart rate of Group A is 9% higher than that of Group A, $t(31) = 1.156$, $p > 0.05$. The RR interval of Group B is 9% lower than that of Group A, $t(31) = -1.133$, $p > 0.05$. Regarding the LF/HF ratio, the average value of Group A is 5% higher than that of Group B, $t(31) = 1.182$, $p > 0.05$.

Figure 9 captures the difference between the driving phase and the relaxation phase in the HRV. The driving task influences stress similarly in both groups. The result of the *t*-test for the three stress measurements indicates that there are no significant differences. In both groups, driving caused an increase in stress indicators.

**Table 12.** Spanish drivers grouped by psychoticism score.

| Group | Number of Drivers | Psychoticism Thresholds | Average Psychoticism Score |
|---|---|---|---|
| A | 7 | [2–6] | 2.00 ± 0.00 |
| B | 26 | [0–1] | 0.69 ± 0.47 |

**Table 13.** HRV according to the psychoticism score obtained by Spanish drivers.

| Measurement | Phase | Group | Average Value |
|---|---|---|---|
| Heart Rate | Relaxation | A | 80.1 bpm $\pm$ 15.3 bpm |
| | | B | 74.0 bpm $\pm$ 14.9 bpm |
| | Driving | A | 84.9 bpm $\pm$ 12.8 bpm |
| | | B | 78.0 bpm $\pm$ 14.3 bpm |
| RR Intervals | Relaxation | A | 771.0 ms $\pm$ 139.9 ms |
| | | B | 842.8 ms $\pm$ 172.8 ms |
| | Driving | A | 722.9 ms $\pm$ 111.9 ms |
| | | B | 795.3 ms $\pm$ 158.0 ms |
| LF/HF ratio | Relaxation | A | 5.1 $\pm$ 4.6 |
| | | B | 3.1 $\pm$ 2.0 |
| | Driving | A | 5.7 $\pm$ 3.7 |
| | | B | 4.2 $\pm$ 2.9 |

The HRV of Spanish drivers, according to their psychoticism score, is divided into two phases: relaxation and driving. Each phase is further divided into two subgroups, [A] and [B], and grouped in the table by grey or white blocks.

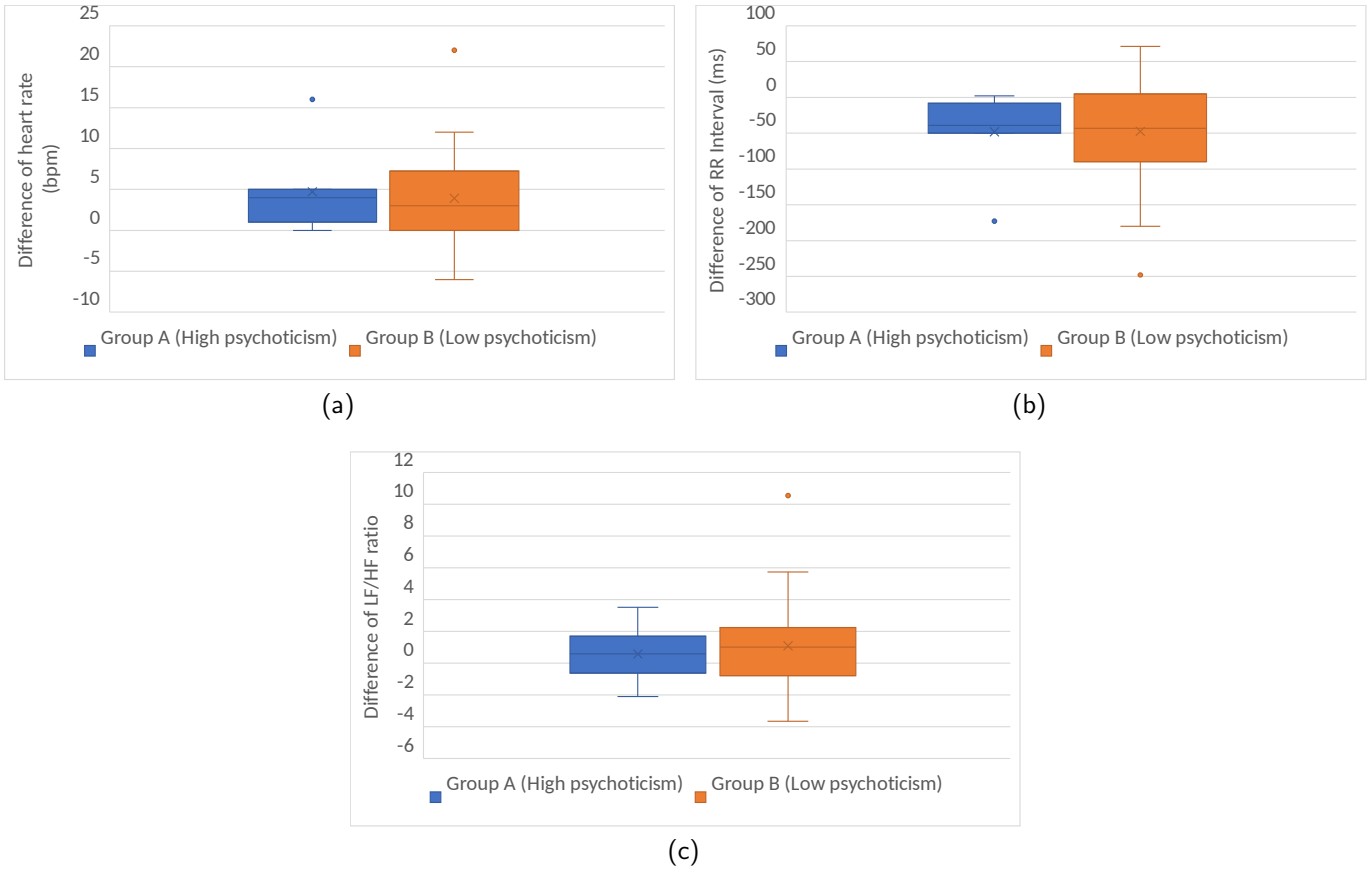

**Figure 9.** Difference of HRV between the driving and relaxation phase according to the psychoticism level in Spain. (**a**) Heart Rate; (**b**) RR Interval; (**c**) LF/HF ratio.

## 5. Discussion of the Results

This section provides a comparative analysis of the results obtained in Germany and Spain. Cultural differences between the countries were studied by Hofstede [43]. In his study, the analysis of national cultures was based on several dimensions, which characterize an important aspect of the culture. According to Hofstede, there are significant cultural differences between Spain and Germany, especially in the dimensions of collectivism, power

distance, uncertainty avoidance and individualism. Each of the dimensions can be associated with personality traits like extroversion, neuroticism and psychoticism. According to Hofstede and comparing the power distance index of Spain (56) and Germany (35) the results suggest that Spaniards tend to be more collectivist, hierarchical, and risk-averse than Germans.

Additionally, in numerous studies, Neuroticism, a personality trait linked to emotional instability, anxiety, and chronic worry, has been linked to elevated stress levels and an increased likelihood of committing driving errors. For Spanish drivers, a high score on this personality trait has been consistently linked to elevated stress levels, both during the relaxation phase of the experiment and while driving. For German drivers, the influence on stress metrics appeared to be less. The only significant distinctions between low-scoring and high-scoring participants were found in the LF/HF ratio measured while driving. This could be attributed to underlying mental health issues. Spain ranks among the top nations in terms of anxiolytic consumption [44]. In addition, the region (Asturias, Spain) where the study was conducted has one of the highest mortality rates due to suicide, 16.5 persons per 100,000 inhabitants [45].

Extroversion/introversion is a personality trait that is characterized by the focus of interest, which can be outside or inside the subject. People with a high score in extroversion are more sociable, lively and more aware of what is happening in the environment around them. Extroverted people are more prone to accidents in monotonous conditions. In our study, as with neuroticism, we can see that this personality trait has a greater effect in Spain. During the relaxation phase, we found significant differences in the LF/HF ratio between introverted and extroverted Spanish drivers. The drivers with the highest score in extroversion obtained a higher value in this measure, which implies greater stress. This is in line with what was observed by other authors in the literature, where extroverted drivers feel the need to seek sensations in monotonous scenarios [39,46]. Regarding the driving phase, we see that this task has a significant impact on the stress level of Spanish introverted drivers. In the case of Germany, in the relaxation phase, extroverted drivers suffered greater stress on average. In the driving phase, the most stressed drivers were the introverted drivers. However, in both phases, the differences were not significant.

Psychoticism is a personality trait characterized by aggression, impulsivity and egocentrism. A high score on this dimension is correlated with a higher risk of traffic accidents, with sensation seeking and a worse disposition to establish positive social relationships with other road users [47]. In our study, we found no significant differences in the level of stress according to this personality trait in either country. This could be due to the fact that none of the participants in the study presented values higher than three in this dimension.

The work initially postulated two hypotheses:

- it suggests a significant relationship between personality traits and driving stress, aiming to investigate whether personality traits have a notable influence on the experienced stress while driving.
- it states that the degree to which personality traits influence driving stress depends on cultural differences between Spain and Germany. Aiming to explore the potential influencing effects of cultural differences on the relationship between driving stress and personality traits.

The research findings partially supported both hypotheses. Analysis of the collected data indicated that individuals with higher levels of neuroticism exhibited elevated stress levels in both Germany and Spain (Tables 3 and 9). Additionally, tendencies related to extroversion/introversion were similar in Germany and Spain. Upon examining Tables 5 and 11, it was found that individuals classified as extroverts in both countries were more prone to stress during relaxation. In line with the second hypothesis, we observed slight variations in behavior between Germans and Spaniards in certain cases. Specifically, within the group of introverts in Germany, we noticed a slightly higher level of stress while driving compared to their counterparts in Spain. It should be noted that the difference between the two groups in Germany is very subtle.

For future works, it may be beneficial to consider additional parameters. The simulation software has the capability to store driving behavior data, such as steering, acceleration, braking, and traffic infractions, which could enhance the accuracy of measurements. This detailed driving record could provide a more nuanced understanding of participants' stress relationships. Additionally, validating the fund relations could be further enhanced by comparing them with real-world data. Aligning simulated driving scenarios with real-world situations would strengthen the reliability and ecological validity of the research outcomes. Validation of the simulator with real-world data would contribute to the robustness and practical relevance of the research.

## 6. Conclusions

In this work, we have analyzed and compared the stress and the personality of a group of Spanish region Asturias and German Lake Constance Region drivers. Both groups were analyzed by comparing the relaxed and driving states and their personalities. In this work, we focused on the personality groups neuroticism, extroversion and psychoticism. These traits are strongly associated with a high probability of making driving errors and suffering from more stress. This study showed some differences in the stress scores between Spain and Germany, although the populations were in a similar range. The most significant differences were primarily observed in the low-scoring and high-scoring drivers in the LF/HF ratio obtained while driving by neuroticism and extroversion/introversion personality. We could not observe a significant difference in the stress level by psychoticism personality. This behavior could be explained with the fact that none of the participants from Germany or Spain in the presented study could be categorized as psychotic. One possible way to improve the results is to use a much more potent stressor. As a possible novel stressor, virtual reality can be used. However, virtual reality as a stressor will need to be analyzed separately, and additional filtering of the bio-vital data of the participants would be needed. Additional validation of the results could be acquired by using a cortisol test using saliva samples. This analysis could show a more precise and substantial relation between stressors, stress, and personality traits while driving. As mentioned in the discussion section, we could observe a difference between the groups and the personalities. This personality plays a significant role in the amount of stress a person suffers from. In a future version of the work, virtual reality can be used as a more robust stressor due to the effects of immersion in the simulation and the lack of motion feedback. An alternative option to detect and increment the sampling size is using an explainable machine learning algorithm for detection and generative adversarial networks (GANS) for generating additional data sets. The findings of this research can contribute to enhancing safety in various domains where human-machine interactions are crucial or where human decision-making is susceptible to personality and stress factors. This extends beyond driving a car and could be applicable in sectors such as medicine, business decision-making, or trading, where a single decision can have severe outcomes. In such applications, providing feedback to the decision-maker is vital, allowing for the reconsideration and reevaluation of decisions. For example, the findings from this research can help in the further development and fine-tuning of advanced driver assistance systems, considering cultural factors as influential elements in understanding and forecasting driver behavior, thereby preventing traffic accidents. Additionally, applications for semi-automated driving are conceivable, where system control is context-dependent, and in emergencies, an override mechanism can be implemented.

**Author Contributions:** Conceptualization, W.D.S. and V.C.; methodology, W.D.S. and V.C.; software, W.D.S. and V.C.; validation, W.D.S. and V.C.; formal analysis, J.A.O., W.D.S. and V.C.; investigation, J.A.O., D.M., W.D.S. and V.C.; resources, J.A.O. and R.S.; data curation, W.D.S. and V.C.; writing—original draft preparation, D.M., W.D.S. and V.C.; writing—review and editing, J.A.O., W.D.S. and V.C.; visualization, W.D.S. and V.C.; supervision, R.S., J.A.O. and N.M.M.; project administration, W.D.S.; funding acquisition, J.A.O., R.S. and W.D.S. All authors have read and agreed to the published version of the manuscript.

**Funding:** This publication was supported by the Spanish National Research Program under Project TIN2017-82928-R and has been partially supported by the "Generation of Reliable Synthetic Health Data for Federated Learning in Secure Data Spaces" Grant PID 2022-141045OB-C42 funded by MCIN/ AEI/10.13039/501100011033 and by "ERDF A way of making Europe" by the "European Union".

**Informed Consent Statement:** Informed consent was obtained from all subjects involved in the study.

**Data Availability Statement:** The data presented in this study are available on request from the corresponding author. The data are not publicly available due to privacy restrictions, but are available on request for scientific research.

**Conflicts of Interest:** The authors declare no conflict of interest. The funders had no role in the design of the study; in the collection, analyses, or interpretation of data; in the writing of the manuscript; or in the decision to publish the results.

## Abbreviations

The following abbreviations are used in this manuscript:

| | |
|---|---|
| EPQR | Eysenck Personality Questionnaire |
| EPQR-A | Eysenck Personality Questionnaire Revised-Abbreviated |
| HRV | heart rate variability |
| PRV | pulse rate variability |
| ECG | Electro Cardio Gram |

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
