# Peer review of "Analysis of the Relationship between Personality Traits and Driving Stress Using a Non-Intrusive Wearable Device"

_electronics, doi:10.3390/electronics13010159_

Round 1

Reviewer 1 Report

Comments and Suggestions for Authors

Attached ,  you  can  find  my  comments !

Author Response

Dear REVIEWER,
First at all I want to thank you very much for the detailed and nice work you did. I agree whit almost all points you described and I see there still a loof of potencial to improve and grow. 

I added the experiment design as a diagram. 

Comment
There are some limitations to the study in its current form :
    no flowchart of the study;
    2-3 hypotheses are not formulated at the beginning of the study;
    a small number of books/manuals as theoretical background.
Answer
    A chart explaining the experiment has been added. The graphic illustrating the study has been placed in the methodology section.
    The hypotheses were reformulated and written more explicitly.
    Literature and other publications were added. This recommendation greatly improved the quality

Comment

    Alongside other aspects, the study aims to determine the extent to which the personality traits of each sample member correlate with the level of driving stress and cultural background (Germany and Spain).
    1.    I suggest the authors to design a flow chart of the whole study (literature review + research method + selection of the sample of 71 persons + steps taken + results + etc.) , line 68, page 2, and to state 2-3 hypotheses that have been considered by the authors since the design of the research.

    Argument:

    This way of presentation would favour a quicker and more unified understanding of the basic idea of the study; it should highlight the large amount of work done by the authors in the field research phase; if 2-3 hypotheses are formulated at the beginning of the study, then in point 5,
    „Discussion of results”, the authors should show whether those hypotheses are confirmed/refuted

Answer

    This comment was also helpful. I included a brief discussion of the hypothesis in the discussion.

Comment

    2.    The syntagma used by the authors to name section 2 of the study, "State of the art", is not, I think, appropriate for such a study. I recommend the phrase "literature review".

Answer

    The title of section 2 was changed. 

Comment

    3.    In the total of 40 references at the end of the study, only 2 sources are books/volumes; the other
    38 references are small articles/studies. I suggest the authors to include at least 5-6 other books/volumes as references in section 2 of the study, pp. 2-5, where the theoretical background of the article is presented. Such volumes should cover both personality and social behaviour and stress management (as the whole study is designed around these 2 topics).

Answer

    Literature and other publications were added.

Comment

    In my opinion, a better theoretical grounding of the study would subsequently ensure greater credibility for the design, methods and results arrived at by the authors. In addition, such a theoretical grounding should lead to a greater attractiveness of the study, including for the general public, assuming publication of the article. Between various basic volumes/manuals on the 2 topics, I suggest to the authors books such as:

    -    Martin Fishbein, Icek Ayzen - Belief, Attitude, Intention and behaviour, Addison Wesley Publishing Company, 1975.
    -    Icek Ayzen - Attitudes, Personality and Behaviour, Second Edition, McGraw Hill Education, 2005
    -    P.M. Lehrer / R.L.Woolfolk -Principles and Practice of Stress Management , Fourth Ed , Guilford Press , 2021
    -    D.McIntoch et al – Stress. The Psychology of Managing Pressure, DK , 2017

4.    On page 6, line 251-259, I suggest the authors explicitly mention whether the 71 drivers were:
    -    professional;
    -    in the amateur category ( regular citizens with a driving licence);
    -    both categories;
    -    native German and Spanish citizens respectively or citizens from other countries?

Answer

    This additional information has been added. It facilitates comprehension.

Comment

    5.    On page 7, line 310-327, it is not sufficiently clear how exactly the "driving simulator" was linked to the other steps of the "field research" (it can only be inferred that the authors used the simulator to select "narrow streets and unsupervised pedestrian crossings with multi- lane roads and highways"); this is the most likely scenario; to be clarified.

Answer

    The subsection has been restructured to provide a clearer understanding that the driving simulator is a safe method for inducing stress in participants. While real-world data would be preferable, it is associated with additional issues. It is worth to mentin that we are currently in the initial stages of planning to collect real-world data but this still in the future and is not relevant fot this work.

Comment

    6.    Starting on page 9 and following, in all figures 1, 2, 3, 4, 5, 6, up to page 15, the authors should present more clearly/visibly each individual graph (at least enlarged the writing and figures in each graph).

Answer

    The resolution and size of the figures have been enhanced, resulting in improved readability. The images are now in vector format.

Comment

    7.    At the beginning of point 5, „Discussion of the results”, line 482, page 14, the authors should add 3-4 additional sentences showing:
    -    What is the general relationship established by psychologists between the cultural specificity of the two countries (Spain + Germany), even if they did NOT study this aspect directly , and possible differences in personality traits;
    -    What is the relationship between emotional intelligence (EI) and personality patterns in general?

Answer

    Additional information about this was added in the discussion. The relationship between Emotional Intelligence and personality patterns could not be covered in this paper as it goes beyond the scope of the paper and in my personal opinion would be worthy of a separate article.

Comment

    In the final part of the study, page 16, in the conclusions, the authors should indicate the potential beneficiaries of the study in pragmatic terms (e.g. companies in the car industry; companies producing software and other digital equipment; public institutions managing motorways, etc.) as well as in theoretical terms. Also in this part of the study I suggest the authors to add 2-3 sentences, as their personal conclusion, what would be the forward-looking relationship between the basic idea of the study and the "Self driving car" project?

Answer

    Thanks for your feedback. I have added your recommendation to the conclusion. This improves the quality and helps to explain the potential uses in industry and research.

Comment

    2.    There are, however, small formal and/or editing errors (e.g. on page 5, line 233, the numbering 3.0.1. is incorrect as it should start with 3.1. etc.; some references in the total of 40 at the end of the article are incomplete or not uniformly made, such as reference 4, 5, 7, 12 18, 25, 26 etc.).

    Thank you for noticing the small format/editing errors. They have been corrected.

Reviewer 2 Report

Comments and Suggestions for Authors

This paper has analyzed the relationship between personality traits and driving stress using a non-intrusive wearable device. They focused on the investigation of stress and its implication. The topic is interesting and authors have made some valuable contributions. However, I have prepared some extensive comments for the paper.

1.         I suggested authors to double check their complete affiliation including country.

2.         Authors should provide the estimation of stress levels for the selected group of individuals as numerical results at the end of the abstract so that readers can have a deep understanding.

3.         I suggest the authors to improve the introduction section by better highlighting to what extent their study contributes to close a gap in the existing literature and/or practice.

4.         The introduction lacks the necessary information that explains the importance and direction of the research. The paper needs to clarify the motivation, challenges, contribution, objective, significance, and others. All the information should be presented in chronological order.

5.         In the literature review, authors have provided some studies related to their topic, but fails to mention the difference between those studies with their current study. I suggest the authors to make the studies on literature review as a Table that will include, author, method, objective, and main findings. In this way, readers can have a clear picture of the main difference between the current study with previous mentioned ones.

6.         The resolution of writing withing all Figures is very low and not clearly visible. I suggested authors to increase the resolution and make each Figure more presentable. Origin software can better draw the Figure instead of excel.

7.         The discussion is poor. I suggested authors to improve it and compare their findings with previous studies.

8.         Also, the managerial implications should be provided and discussed right after the discussion section.

9.         Conclusion should be reworked. You should mention the most important numerical results briefly, summarize the paper, mention the limitations and also give more precise directions of the future work.

10.     Overall English should be improved. There are some mistakes, you should check entire manuscript for these mistakes.

Comments on the Quality of English Language

Moderate editing English is required

Author Response

Dear Reviewer,
Thank you for the interesting and very useful comments. I have taken almost all the comments into account in my manuscript. I am very grateful for your criticism. 
Below are the answers to your comments.

1.    I suggested authors to double check their complete affiliation including country.
a.    Thank you for the affiliation hint. All affiliations have been updated accordingly.

2.    Authors should provide the estimation of stress levels for the selected group of individuals as numerical results at the end of the abstract so that readers can have a deep understanding.
a.    We have expanded the abstract and included data estimates to make it easier for readers to understand the results of the study. 

3.    I suggest the authors to improve the introduction section by better highlighting to what extent their study contributes to close a gap in the existing literature and/or practice.
a.    Thank you for your suggestion. We have made some restructuring, including adding a clear statement of our hypotheses and how this work can contribute to the field of stress and personality research, as well as potential applications.

4.    The introduction lacks the necessary information that explains the importance and direction of the research. The paper needs to clarify the motivation, challenges, contribution, objective, significance, and others. All the information should be presented in chronological order.
a.    Thank you for your comment. While we considered presenting the information chronologically, we ultimately decided to cluster the topics and emphasize connections between each research. This approach offers a more understandable overview. 

5.    In the literature review, authors have provided some studies related to their topic, but fails to mention the difference between those studies with their current study. I suggest the authors to make the studies on literature review as a Table that will include, author, method, objective, and main findings. In this way, readers can have a clear picture of the main difference between the current study with previous mentioned ones.
a.    Thank you for your comment. In the literature review, we included a table that differentiates the methods and clusters them based on the approach used. We also briefly mentioned what our observations were.

6.    The resolution of writing withing all Figures is very low and not clearly visible. I suggested authors to increase the resolution and make each Figure more presentable. Origin software can better draw the Figure instead of excel.
a.    the resolution and size of the figures have been improved.

7.    The discussion is poor. I suggested authors to improve it and compare their findings with previous studies.
a.    The discussion was expanded. 

8.    Also, the managerial implications should be provided and discussed right after the discussion section.
a.    The modified version of the conclusion mentions areas where systems that include cultural factors for decision-making could be used. If you believe that more precise examples should be mentioned, please do not hesitate to inform me.

9.    Conclusion should be reworked. You should mention the most important numerical results briefly, summarize the paper, mention the limitations and also give more precise directions of the future work.
a.    The recommendations you made were mostly added to the work. 

10.    Overall English should be improved. There are some mistakes, you should check entire manuscript for these mistakes.
a.    The English was checked with Grammarly software. Most mistakes have now been corrected.

Reviewer 3 Report

Comments and Suggestions for Authors

Nice paper, well written. It can be argued that presents some flaws in the methodology in the sense that the situation analyzed could be a little far from the behavior and signals of drivers in real world situations.

Please comment the meaning of a measure reported as : "2.5 ± 2.5" (Table 4)  If the measures are so inaccurate how can you classify stress levels? This one is not an isolataed case.

Please report more details on the measurement instrumenation. Photos, details on Polar H10 chest strap, details about Kubios, datails and images of the simulator.

Figures 1-6 should be made more readable.

Have you considered to analyze the case of real daily situations for commutees? It would be interesting to record data of commutees during they travel from home to work and viceversa. The will to get on time at work and road traffic would be potent sterssors.

Does the simulator allow you to extract data on driving style? Eg: accelerations, steering, intensity of braking..

Comments on the Quality of English Language

English is good to the best of my non-native-english-speaking experience.

Author Response

Dear Reviewer,
First of all, I want to thank you very much for your comments. They help to improve the quality of the work. I have tried to take almost all the comments into account in my manuscript. 
Below are the responses to your comments.

1.    Please comment the meaning of a measure reported as : "2.5 ± 2.5" (Table 4)  If the measures are so inaccurate how can you classify stress levels? This one is not an isolataed case.
a.    This notation indicates that the average value of the LF/HF ratio is 2.5, with a standard deviation of 2.5. This does not mean that this measure is not accurate for classifying stress level, but rather that it reflects the inherent variability in human stress responses. Different individuals can exhibit vastly different physiological responses to stress. For this reason, we have included additional measures for a more comprehensive assessment of stress and we have analyzed the differences observed by the same individual in the relaxation phase and while driving.

2.    Please report more details on the measurement instrumenation. Photos, details on Polar H10 chest strap, details about Kubios, datails and images of the simulator.
a.    More details like foot notes, images and text was added. 

3.    Figures 1-6 should be made more readable.
a.    the resolution and size of the figures have been improved.

4.    Have you considered to analyze the case of real daily situations for commutees? It would be interesting to record data of commutees during they travel from home to work and viceversa. The will to get on time at work and road traffic would be potent sterssors.
a.    We have been considering that option for some time and are planning to organise a study with a local car-sharing company in Germany (and possibly in Spain) in the coming year. The results of this study will be compared with those of the simulation to determine if the tendencies and behaviours are similar. We are currently awaiting approval from the company.

5.    Does the simulator allow you to extract data on driving style? Eg: accelerations, steering, intensity of braking.
a.    Yes, the simulator suprts the collection of data. This was added to the comments in future work. 

Round 2

Reviewer 2 Report

Comments and Suggestions for Authors

The authors have significantly improved the manuscript, thereby I recommend to publish the manuscript